# Spin-Weighted Spherical CNNs

**Carlos Esteves**
GRASP Laboratory
University of Pennsylvania
machc@seas.upenn.edu

**Ameesh Makadia**
Google Research
makadia@google.com

**Kostas Daniilidis**
GRASP Laboratory
University of Pennsylvania
kostas@cis.upenn.edu

## Abstract

Learning equivariant representations is a promising way to reduce sample and model complexity and improve the generalization performance of deep neural networks. The spherical CNNs are successful examples, producing SO(3)-equivariant representations of spherical inputs. There are two main types of spherical CNNs. The first type lifts the inputs to functions on the rotation group SO(3) and applies convolutions on the group, which are computationally expensive since SO(3) has one extra dimension. The second type applies convolutions directly on the sphere, which are limited to zonal (isotropic) filters, and thus have limited expressivity. In this paper, we present a new type of spherical CNN that allows anisotropic filters in an efficient way, without ever leaving the spherical domain. The key idea is to consider spin-weighted spherical functions, which were introduced in physics in the study of gravitational waves. These are complex-valued functions on the sphere whose phases change upon rotation. We define a convolution between spin-weighted functions and build a CNN based on it. The spin-weighted functions can also be interpreted as spherical vector fields, allowing applications to tasks where the inputs or outputs are vector fields. Experiments show that our method outperforms previous methods on tasks like classification of spherical images, classification of 3D shapes and semantic segmentation of spherical panoramas.

## 1 Introduction

Learning representations from data enables a variety of applications that are not possible with other methods. Convolutional neural networks (CNNs) are powerful tools in representation learning, in great part due to their translation equivariance property that allows weight-sharing, exploiting the natural structure of audio, image, or video inputs.

Recently, there has been significant work extending equivariance to other groups of transformations [20, 9, 13, 44, 33, 17, 45, 40, 43, 18, 4] and designing equivariant CNNs on non-Euclidean domains [11, 16, 26, 37, 35, 8, 27, 37, 48]. Successful applications have been demonstrated in tasks such as 3D shape analysis [16, 18], medical imaging [42, 3], satellite/aerial imaging [13, 21], cosmology [13, 35], physics/chemistry [11, 26, 1]. Favorable results were also shown on popular upright natural image datasets such as CIFAR10/100 [39].

Rotation equivariant CNNs are the natural way to learn feature representations on spherical data. There are two prevailing designs, (a) convolution between spherical functions and zonal (isotropic; constant per latitude) filters [16], and (b) convolutions on $\mathbf{SO}(3)$ after lifting spherical functions to the rotation group [11]. There is a clear distinction between these two designs: (a) is more efficient allowing to build representational capacity through deeper networks, and (b) has more expressive filters but is computationally expensive and thus is constrained to shallower networks. The question we consider in this paper is: how can we achieve the expressivity/representation capacity of $\mathbf{SO}(3)$ convolutions with the efficiency and scalability of spherical convolutions?

In this paper, we propose to leverage spin-weighted spherical functions (SWSFs), introduced by Newman and Penrose [34] in the study of gravitational waves. These are complex-valued functions on the sphere that, upon rotation, suffer a phase change besides the usual spherical translation.

Our key observation is that a combination of SWSFs allows more expressive representations than scalar spherical functions, avoiding the need to lift features to the higher dimensional $\mathbf{SO}(3)$. It also enables anisotropic filters, removing the filter constraint of purely spherical CNNs.

We define convolutions and cross-correlations of SWSFs. For bandlimited inputs, the operations can be computed exactly in the spectral domain, and are equivariant to the continuous group $\mathbf{SO}(3)$. We build a CNN where filters and features are sets of SWSFs, and adapt nonlinearities, batch normalization, and pooling layers as necessary.

Besides more expressive and efficient representations, we can interpret the spin-weighted features as equivariant vector fields on the sphere, enabling applications where the inputs or outputs are vector fields. Current spherical CNNs [11, 16, 26, 35] cannot achieve equivariance in this sense, as illustrated in Fig. 1.

To evaluate vector field equivariance, we introduce a variation of MNIST where the images and their gradients are projected to the sphere. We propose three tasks on this dataset: 1) vector field classification, 2) vector field prediction from scalar fields, 3) scalar field prediction from vector fields. We also evaluate our model on spherical image classification, 3D shape classification, and semantic segmentation of spherical panoramas.

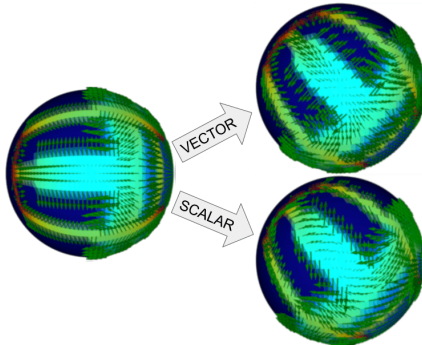

Figure 1: Colors represent a scalar field, and the green vectors represent a vector field. Upon rotation, scalar fields transform by simply moving values to another position, while vector fields move and also rotate. Treating vector fields as multi-channel scalars (bottom-right) results in incorrect behavior. The spin-weighted spherical CNNs equivariantly handle vector fields as inputs or outputs.

To summarize our contributions,

1. We define the convolution and cross-correlation between sets of spin-weighted spherical functions. These are $\mathbf{SO}(3)$ equivariant operations that respect the SWSFs properties.

2. We build a CNN based on these operations and adapt usual CNN components for sets of SWSFs as features and filters. This is, to the best of our knowledge, the first spherical CNN that operates on vector fields.

3. We demonstrate the efficacy of the spin-weighted spherical CNNs (SWSCNNs) on a variety of tasks including spherical image and vector field classification, predicting vector field from images and conversely, 3D shape classification and spherical image segmentation.

4. We will make our code and datasets publicly available at `https://github.com/daniilidis-group/swscnn`.

## 2   Related work

**Equivariant CNNs**   The first equivariant CNNs were applied to images on the plane [20, 13]. Cohen and Welling [9] formalized these models and named them group equivariant convolutional neural networks (G-CNNs). While initial methods were constrained to small discrete groups of rotations on the plane, they were later extended to larger groups [41], continuous rotations [44], rotations and scale [17], 3D rotations of voxel grids [43, 40], and point clouds [37].

**Spherical CNNs**   G-CNNs can be extended to homogeneous spaces of groups of symmetries [28]; the quintessential example is the sphere $S^2$ as a homogeneous space of the group $\mathbf{SO}(3)$, the setting of spherical CNNs. There are two main branches. The first branch, introduced by Cohen et al. [11], lifts the spherical inputs to functions on $\mathbf{SO}(3)$, and its filters and features are functions on the group $\mathbf{SO}(3)$, which is higher dimensional and thus more computationally expensive to process. Kondor et al. [26] is another example. The second branch, introduced by Esteves et al. [16], is purely spherical and has filters and features on $S^2$, using spherical convolution as the main operation. In this case, the

filters are constrained to be zonal (isotropic), which limits the representational power. Perraudin et al. [35] also uses isotropic filters, but with graph convolutions instead of spherical convolutions.

Our approach lies between these two branches. It is not restricted to isotropic filters but it does not have to lift features to $\mathbf{SO}(3)$; we employ sets of SWSFs as filters and features.

A separate line of work developed spherical CNNs that are not rotation-equivariant [24, 47], which rely on the strong assumption that the inputs are aligned.

**Equivariant vector fields** Our approach can equivariantly handle spherical vector fields as inputs or outputs. Marcos et al. [33] introduced a planar CNN whose features are vector fields obtained from rotated filters. Cohen and Welling [12] formalized the concept of feature types that are vectors in a group representation space. This was extended to 3D Euclidean space by Weiler et al. [40]. Worrall et al. [44] introduced complex-valued features on $\mathbb{R}^2$ whose phases change upon rotation; this is similar in spirit to our method, but our features live on the sphere, requiring different machinery.

Cohen et al. [8] introduced a framework that produces vector field features on general manifolds; it was specialized to the sphere by Kicanaoglu et al. [25]. The major differences are that our implementation is fully spectral and we demonstrate it on tasks requiring vector field equivariance. Cohen et al. [10] alluded to the possibility of building spherical CNNs that can process vector fields; we materialize these networks.

## 3 Background

In this section, we provide the mathematical background that guides our contributions. We first introduce the more commonly encountered spherical harmonics, then the generalization to the spin-weighted spherical harmonics (SWSHs). We also describe convolutions between spherical functions, which we will later generalize to convolutions between spin-weighted functions.

**Spherical Harmonics** The spherical harmonics $Y_m^\ell \colon S^2 \to \mathbb{C}$ form an orthonormal basis for the space $L^2(S^2)$ of square integrable functions on the sphere. Any function $f \colon S^2 \to \mathbb{C}$ in $L^2(S^2)$ can be decomposed in this basis via the spherical Fourier transform (SFT) (Eq. (1)), and synthesized back exactly via its inverse (Eq. (2)),

$$\hat{f}_m^\ell = \int_{S^2} f(x)\overline{Y_m^\ell}(x)\, dx, \qquad (1) \qquad f(x) = \sum_{\ell=0}^{\infty} \sum_{|m|\leq \ell} \hat{f}_m^\ell Y_m^\ell(x). \qquad (2)$$

We interchangeably use latitudes and longitudes $(\theta, \phi)$ or points $x \in \mathbb{R}^3$, $\|x\| = 1$ to index the sphere, and we use the hat to denote Fourier coefficients. A function has bandwidth $B$ when only components of order $\ell \leq B$ appear in the expansion.

The spherical harmonics are related to irreducible representations of the group $\mathbf{SO}(3)$ as follows,

$$D_{m,0}^\ell(\alpha, \beta, \gamma) = \sqrt{\frac{4\pi}{2\ell + 1}} \overline{Y_m^\ell(\beta, \alpha)}, \qquad (3)$$

where $\alpha$, $\beta$ and $\gamma$ are ZYZ Euler angles and $D^\ell$ is a Wigner-D matrix.[1] Since $D^\ell$ is a group representation and hence a group homomorphism, we obtain a rotation formula,

$$Y_m^\ell(gx) = \sum_{n=-\ell}^{\ell} \overline{D_{m,n}^\ell(g)} Y_n^\ell(x), \qquad (4)$$

where we interchangeably use an element $g \in \mathbf{SO}(3)$ or Euler angles $\alpha$, $\beta$ and $\gamma$ to refer to rotations.

Consider the rotation of a function represented by its coefficients by combining Eqs. (2) and (4),

$$f(gx) = \sum_{\ell=0}^{\infty} \sum_{n=-\ell}^{\ell} \left( \sum_{m=-\ell}^{\ell} \hat{f}_m^\ell \overline{D_{m,n}^\ell(g)} \right) Y_n^\ell(x). \qquad (5)$$

This shows that when $f(x) \mapsto f(gx)$, its Fourier coefficients transform as

$$\hat{f}_n^\ell \mapsto \sum_m \overline{D_{m,n}^\ell(g)} \hat{f}_m^\ell \qquad (6)$$

Finally, we recall how convolutions and cross-correlations of spherical functions are computed in the spectral domain. Esteves et al. [16] define the convolution between two spherical functions $f$ and $k$ as Eq. (7) while Makadia and Daniilidis [32] and Cohen et al. [11] define the spherical cross-correlation as Eq. (8),

$$(\widehat{k * f})_m^\ell = 2\pi \sqrt{\frac{4\pi}{2\ell + 1}} \hat{f}_m^\ell \hat{k}_0^\ell, \qquad (7) \qquad\qquad (\widehat{k \star f})_{m,n}^\ell = \hat{f}_m^\ell \overline{\hat{k}_n^\ell}, \qquad (8)$$

Both are shown to be equivariant through Eq. (6). The left-hand side of Eq. (7) correspond to the Fourier coefficients of a spherical function, while the left-hand side of Eq. (8) correspond to the Fourier coefficients of a function on $\mathbf{SO}(3)$.

This section laid the foundation for the spin-weighted generalization. We refer to Esteves [15] for a longer exposition on this topic and to Vilenkin and Klimyk [38] and Folland [19] for the full details.

**Spin-Weighted Spherical Harmonics** The spin-weighted spherical functions (SWSFs) are complex-valued functions on the sphere whose phases change upon rotation. They have different types determined by the spin weight.

Let $_s f \colon S^2 \to \mathbb{C}$ be a SWSF with spin weight $s$, $\lambda_\alpha$ a rotation by $\alpha$ around the polar axis, and $\nu$ the north pole. In a conventional spherical function, $\nu$ is fixed by the rotation, so $(\lambda_\alpha(f))(\nu) = f(\nu)$. In a spin-weighted function, however, the rotation results in a phase change,

$$(\lambda_\alpha(_s f))(\nu) = {_s f}(\nu) e^{-is\alpha}. \qquad (9)$$

If the spin weight is $s = 0$, this is equivalent to the conventional spherical functions.

The spin-weighted spherical harmonics (SWSHs) form a basis of the space of square-integrable spin-weighted spherical functions; for all square-integrable $_s f$, we can write

$$_s f(\theta, \phi) = \sum_{\ell \in \mathbb{N}} \sum_{m=-\ell}^{\ell} {_s Y_m^\ell}(\theta, \phi) {_s \hat{f}_m^\ell}, \qquad (10)$$

where $_s \hat{f}_m^\ell$ are the expansion coefficients, and the decomposition is defined similarly to Eq. (1). For $s = 0$, the SWSHs are exactly the spherical harmonics; we have $_0 Y_m^\ell = Y_m^\ell$.

The SWSHs are related to the matrix elements $D_{mn}^\ell$ of $\mathbf{SO}(3)$ representations as follows,

$$D_{m,-s}^\ell(\alpha, \beta, \gamma) = (-1)^s \sqrt{\frac{4\pi}{2\ell + 1}} {_s \overline{Y_m^\ell(\beta, \alpha)}} e^{-is\gamma}. \qquad (11)$$

Note how different spin-weights are related to different columns of $D^\ell$, while the standard spherical harmonics are related to a single column as in Eq. (3). This shows that the SWSHs can be seen as functions on $\mathbf{SO}(3)$ with sparse spectrum, a point of view that is advocated by Boyle [6].

The SWSHs do not transform among themselves upon rotation as the spherical harmonics (Eq. (4)) due to the extra phase change. Fortunately, the coefficients of expansion of a SWSF into the SWSHs do transform among themselves according to Eq. (6). When $_s f(x) \mapsto {_s f}(gx)$,

$$_s \hat{f}_n^\ell \mapsto \sum_m \overline{D_{m,n}^\ell(g)} {_s \hat{f}_m^\ell}. \qquad (12)$$

This is crucial for defining equivariant convolutions between combinations of SWSFs as we will do in Section 4.1. We refer to Castillo [7] and Boyle [5, 6] for more details about SWSFs.

## 4 Method

We introduce a fully convolutional network, the spin-weighted spherical CNN (SWSCNN), where layers are based on spin-weighted convolutions, and filters and features are combinations of SWSFs. We define spin-weighted convolutions and cross-correlations, show how to efficiently implement them, and adapt common neural network layers to work with combinations of SWSFs.

## 4.1 Spin-Weighted Convolutions and Cross-Correlations

We define and evaluate the convolutions and cross-correlations in the spectral domain. Consider a set of spin weights $W_F$, $W_K$ and sets of functions $F = \{{}_sf\colon S^2 \to \mathbb{C} \mid s \in W_F\}$ and filters $K = \{{}_sk\colon S^2 \to \mathbb{C} \mid s \in W_K\}$ to be convolved.

**Spin-weighted convolution**   We define the convolution between $F$ and $K$ as follows,

$$ {}_s(\widehat{F * K})^\ell_m = \sum_{i \in W_F} {}_i\hat{f}^\ell_m \, {}_s\hat{k}^\ell_i, \tag{13} $$

where $s \in W_K$ and $-\ell \leq m \leq \ell$. Only coefficients ${}_s\hat{k}^\ell_i$ where $i \in W_F$ influence the output, imposing sparsity in the spectra of $K$. The convolution $F * K$ is also a set of SWSFs with $s \in W_K$, the same spin weights as $K$; we leverage this to specify the desired sets of spins at each layer.

We show this operation is $\mathbf{SO}(3)$ equivariant by applying the rotation formula from Eq. (12). Let $\lambda_g$ denote a rotation of each ${}_sf(x) \in F$ by $g \in \mathbf{SO}(3)$. We have,

$$
\begin{aligned}
{}_s(\widehat{\lambda_g F * K})^\ell_n &= \sum_{i \in W_F} \sum_m \overline{D^\ell_{m,n}(g)} \, {}_i\hat{f}^\ell_m \, {}_s\hat{k}^\ell_i \\
&= \sum_m \overline{D^\ell_{m,n}(g)} \sum_{i \in W_F} {}_i\hat{f}^\ell_m \, {}_s\hat{k}^\ell_i \\
&= \sum_m \overline{D^\ell_{m,n}(g)} \, {}_s(\widehat{F * K})^\ell_m \\
&= \lambda_g({}_s(\widehat{F * K})^\ell_n).
\end{aligned}
\tag{14}
$$

Now consider the spherical convolution defined in Eq. (7). It follows immediately that it is, up to a constant, a special case of the spin-weighted convolution, where $F$ and $K$ have only one element with $s = 0$, and only the filter coefficients of form ${}_0\hat{k}^\ell_0$ are used.

**Spin-weighted cross-correlation**   We define the cross-correlation between $F$ and $K$ as follows,

$$ {}_s(\widehat{F \star K})^\ell_m = \sum_{i \in W_F \cap W_K} {}_i\hat{f}^\ell_m \, \overline{{}_i\hat{k}^\ell_s}, \tag{15} $$

In this case, only the spins that are common to $F$ and $K$ are used, but all spins may appear in the output, so it can be seen as a function on $\mathbf{SO}(3)$ with dense spectrum. To ensure a desired set of spins in $F \star K$, we can sparsify the spectra in $K$ by eliminating some orders. A procedure similar to Eq. (14) proves the $\mathbf{SO}(3)$ equivariance of this operation.

The spin-weighted cross-correlation generalizes the spherical cross-correlation. When $F$ and $K$ contain only a single spin weight $s = 0$, the summation in Eq. (15) will contain only one term and we recover the spherical cross-correlation defined in Eq. (8).

**Examples**   To visualize the convolution and cross-correlations, we use the phase of the complex numbers and define local frames to obtain a vector field. We visualize combinations of SWSFs by associating pixel intensities with the spin-weight $s = 0$ and plotting vector fields for each $s > 0$.

Consider an input $F = \{{}_0f, {}_1f\}$ and filter $K = \{{}_0k, {}_1k\}$, both with spin weights 0 and 1. Their convolution also has spins 0 and 1, as shown on the left side of Fig. 2. Now consider a scalar valued (spin $s = 0$) input $F = \{{}_0f\}$ and filter $K = \{{}_0k\}$. The cross-correlation will have components of every spin, but we only take spin weights 0 and 1 to visualize (this is equivalent to eliminating all orders larger than 1 in the spectrum of $k$); Fig. 2 shows the results.

## 4.2 Spin-weighted spherical CNNs

Our main operation is the convolution defined in Section 4.1. Since components with the same spin can be added, the generalization to multiple channels is immediate. The convolution combines features of different spins, so we enforce the same number of channels per spin per layer. Each feature map then consists of a set of SWSFs of different spins, $F = \{{}_s\bar{f}\colon S^2 \to \mathbb{C}^k \mid s \in W_F\}$, where $k$ is the number of channels and $W_F$ the set of spin weights.

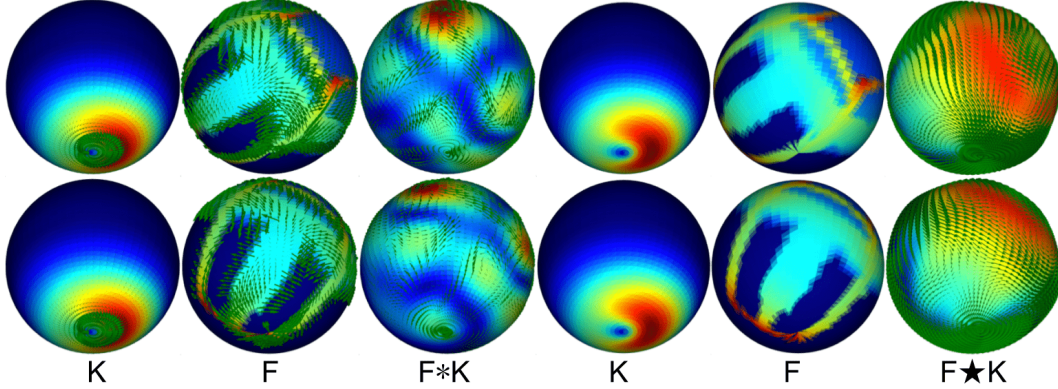

| K | F | F∗K | K | F | F★K |

Figure 2: Left block (2 × 3): convolution between sets of functions of spins 0 and 1. The operation is equivariant as a vector field and outputs carry the same spins. Right block (2 × 3): spin-weighted cross-correlation between scalar spherical functions. The operation is also equivariant and we show outputs corresponding to spins 0 and 1. The second row shows the effect of rotating the input $F$.

**Filter localization**   We compute the convolutions in the spectral domain but apply nonlinearities, batch normalization and pooling in the spatial domain. This requires expanding the feature maps into the SWSHs basis and back at every layer, but the filters themselves are parameterized by their spectrum. We follow the idea of Esteves et al. [16] to enforce filter localization with spectral smoothness. Their filters are of the form ${}_0\hat{k}_0^\ell$, so the spectrum is 1D and can be interpolated from a few anchor points, smoothing it out and reducing the number of parameters. In our case, the filters take the general form ${}_s\hat{k}_m^\ell$ where $s \in W_{F∗K}$ are the output spin weights and $m \in W_F$ are the input spin weights. We then interpolate the spectrum of each component along the degrees $\ell$, resulting in a factor of $|W_{F∗K}||W_F|$ more parameters per layer.

**Batch normalization and nonlinearity**   We force features with spin weight $s = 0$ to be real by taking their real part after every convolution. Then we can use the common rectified linear unit (ReLU) as the nonlinearity and the standard batch normalization from Ioffe and Szegedy [23].

For $s > 0$, we have complex-valued feature maps. Since values move and change phase upon rotation, equivariant operations must commute with this behavior. Pointwise operations on magnitudes satisfy this requirement. Similarly to Worrall et al. [44], we employ a variation of the ReLU to the complex values $z = ae^{i\theta}$ as follows, where $a \in \mathbb{R}^+$ and $b \in \mathbb{R}$ is a learnable scalar.

$$z \mapsto \max(a + b, 0)e^{i\theta}. \tag{16}$$

Batch normalization is also applied pointwise, but it does not commute with spin-weighted rotations because of the mean subtraction and offset addition steps. We adapt it by removing these steps, where $\sigma^2$ is the channel variance, $\gamma \in \mathbb{C}$ is a learnable factor and $\epsilon \in \mathbb{R}^+$ is a constant added for stability,

$$z \mapsto \frac{z}{\sqrt{\sigma^2 + \epsilon}}\gamma. \tag{17}$$

As usual, the variance is computed along the batch during training and along the whole dataset during inference. The variance of a set of complex numbers is real and only depends on their magnitudes; we use a spherical quadrature rule to compute it.

**Complexity analysis**   We follow Huffenberger and Wandelt [22] for the spin-weighted spherical Fourier transform (SWSFT) implementation (see appendix for details), whose complexity for bandwidth $B$ is $\mathcal{O}(B^3)$. While it is asymptotically slower than the $\mathcal{O}(B^2 \log^2 B)$ of the standard SFT from Driscoll and Healy [14], the difference is small for bandwidths typically needed in practice [11, 16, 26]. The rotation group Fourier transform (SOFT) implementation from Kostelec and Rockmore [29] is $\mathcal{O}(B^4)$. Our final model requires $|W|$ transforms per layer, so it is asymptotically a factor $|W|B/\log^2 B$ slower than using SFT as in Esteves et al. [16], and a factor $B/|W|$ faster than using the SOFT as in Cohen et al. [11]. Typical values in our experiments are $B = 32$ and $|W| = 2$.

# 5 Experiments

We start with experiments on image and vector field classification, image prediction from a vector field, and vector field from an image, where all images and vector fields are on the sphere. Next, we show applications to 3D shape classification and semantic segmentation of spherical panoramas.

All experiments use spin weights $0$ and $1$. When inputs do not have both spins, the first layer is designed such that its outputs have. All following layers and filters also have spins $0$ and $1$.

Every model is trained with different random seeds five times and averages and standard deviations (within parenthesis) are reported. See the appendix for training procedure details.

## 5.1 Spherical Image Classification

Our first experiment is on the Spherical MNIST dataset introduced by Cohen et al. [11]. This is an image classification task where the handwritten digits from MNIST are projected on the sphere. Three modes are evaluated depending on whether the training/test set are rotated (R) or not (NR).

We simplify the architecture in Esteves et al. [16] to have a single branch, switch from spherical to spin-weighted convolutions, and adapt the numbers of channels and parameters per filter to match the parameter counts. Table 1 shows the results; we outperform previous spherical CNNs in every mode.

Table 1: Spherical MNIST results. Our model is more expressive than the isotropic and more efficient than the previous anisotropic spherical CNNs, allowing deeper models and improved performance.

|                      | NR/NR     | R/R        | NR/R       | params |
|----------------------|-----------|------------|------------|--------|
| Planar CNN           | 99.07(4)  | 81.07(63)  | 17.23(71)  | 59k    |
| Cohen et al. [11]    | 95.59     | 94.62      | 93.40      | 58k    |
| Kondor et al. [26]   | 96.40     | 96.60      | 96.00      | -      |
| Esteves et al. [16]  | 98.75(8)  | 98.71(5)   | 98.08(24)  | 57k    |
| Ours                 | 99.37(5)  | 99.37(1)   | 99.08(12)  | 58k    |

## 5.2 Spherical Vector Field Classification

One crucial advantage of the SWSCNNs is that they are equivariant as vector fields. To demonstrate this, we introduce a spherical vector field dataset. We start from MNIST [31], compute the image gradients with Sobel kernels and project the vectors to the sphere. To increase the challenge, we follow Larochelle et al. [30] and swap the train and test sets so there are $10\,\mathrm{k}$ images for training and $50\,\mathrm{k}$ for test. We call this dataset the spherical vector field MNIST (SVFMNIST). The vector field is converted to a spin weight $s = 1$ complex-valued function using a predefined local tangent frame per point on the sphere. The inverse procedure converts $s = 1$ features to output vector fields.

Table 2: Spherical vector field MNIST classification results. When vector field equivariance is required, the gap between our method and the spherical and planar baselines is even larger.

|         | NR/NR    | R/R       | NR/R      |
|---------|----------|-----------|-----------|
| Planar  | 97.7(2)  | 50.0(8)   | 14.6(9)   |
| [16]    | 98.4(1)  | 94.5(5)   | 24.8(8)   |
| Ours    | 98.2(1)  | 97.8(2)   | 98.2(7)   |

The first task we consider is classification. We use the same architecture as in the previous experiment, the only difference is that now the first layer maps from spin 1 to spins 0 and 1. Table 2 shows the results. The planar and spherical CNN models take the vector field as a 2-channel input. The NR/R column clearly shows the advantage of vector field equivariance; the baselines cannot generalize to unseen vector field rotations, even when they are equivariant in the scalar sense as [16].

## 5.3 Spherical Vector Field Prediction

The SWSCNNs can also be used for dense prediction. We introduce two new tasks on SVFMNIST, 1) predicting a vector field from an image and 2) predicting an image from a vector field. For these tasks, we implement a fully convolutional U-Net architecture [36] with spin-weighted convolutions.

When the image is a grayscale digit and the vector field comes from its gradients, both tasks can be easily solved via discrete integration and differentiation. We call this case "easy" and show it on the left side of table Table 3. It highlights a limitation of isotropic spherical CNNs; the results show that the constrained filters cannot approximate a simple image gradient operator.

We also experiment with a more challenging scenario, where the digits are colored and the vector fields are rotated based on the digit category. These are semantic tasks that require the network to implicitly classify the input in order to correctly predict output color and vector directions.

Table 3 shows the results. While the planar baseline does well in the "easy" tasks that can be solved with simple linear operators, our model still outperforms it when generalization to unseen rotations is demanded (NR/R). In the "hard" task, the SWSCNNs are clearly superior by large margins. We show sample inputs and outputs in Fig. 3; see the appendix for more.

Table 3: Vector field to image and image to vector field results on SVFMNIST. The SWSCNNs show superior performance, especially on the more challenging tasks. The metric is the mean-squared error $\times 10^3$ (lower is better). All models have around 112k parameters.

| | easy | | | hard | | |
|---|---|---|---|---|---|---|
| | NR/NR | R/R | NR/R | NR/NR | R/R | NR/R |
| **Image to Vector Field** | | | | | | |
| Planar | 0.3(1) | 5.0(1) | 9.3(1) | 16.9(5) | 26.0(1) | 32.9(2) |
| Esteves et al. [16] | 9.7(3) | 31.0(2) | 45.6(7) | 13.3(6) | 28.5(4) | 41.6(4) |
| Ours | 2.9(2) | 3.4(1) | 4.3(1) | 11.6(6) | 9.2(4) | 10.2(6) |
| **Vector Field to Image** | | | | | | |
| Planar | 1.4(1) | 3.2(1) | 6.9(4) | 3.3(2) | 13.4(2) | 21.1(3) |
| Esteves et al. [16] | 3.8(1) | 4.9(2) | 15(2) | 2.6(1) | 6.4(2) | 20.3(9) |
| Ours | 3.5(1) | 3.8(1) | 4.0(1) | 2.6(1) | 2.7(1) | 2.9(1) |

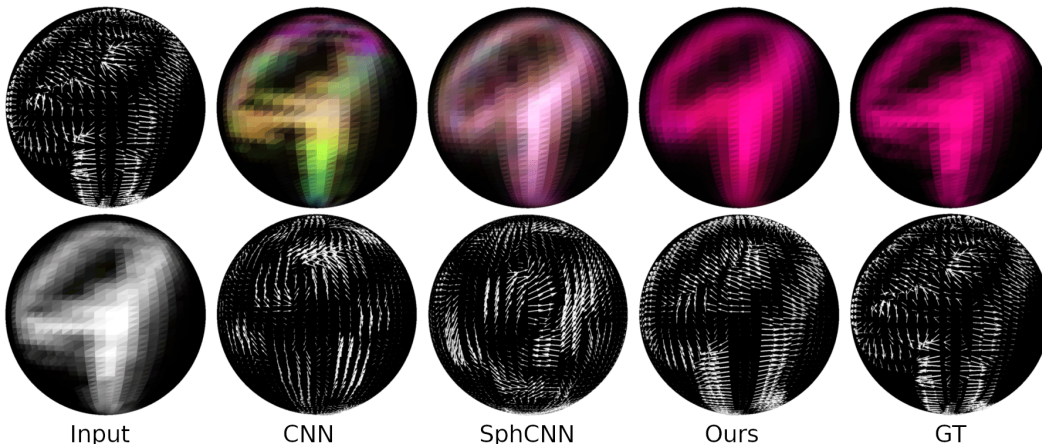

| Input | CNN | SphCNN | Ours | GT |

Figure 3: Input, output and ground truth for dense prediction tasks on rotated train and test sets (R/R). **Top:** vector field to image. Conventional and spherical CNNs [16] predict the incorrect color, in contrast with our SWSCNNs. **Bottom:** image to vector field. Our method predicts the position and orientation of each vector correctly, while the alternatives cannot.

## 5.4  Classification of 3D shapes

We tackle 3D object classification on ModelNet40 [46], following the protocol from Esteves et al. [16] which considers azimuthally and arbitrarily rotated shapes.

Besides more expressive filters, our method also represents the shapes more faithfully on the sphere. Esteves et al. [16] and Cohen et al. [11] cast rays from the shape's center and assign the intersection distance and angle between normal and ray to points on the sphere. Normals are not uniquely determined by a single angle but this limitation was necessary to preserve equivariance as a scalar field.

Table 4: ModelNet40 shape classification accuracy [%]. Our model outperforms previous spherical CNNs while requiring small input size and low parameter count.

|  | upright | rotated |
|---|---|---|
| UGSCNN [24] | 87.3(3) | 81.9(9) |
| SphCNN [16] | 89.3(5) | 88.4(3) |
| Ours | 89.6(3) | 88.8(1) |
| Ours + BE | 90.1(3) | 88.2(2) |

By using SWSCNNs, we can represent any normal direction uniquely, without breaking equivariance. We split the vector in radial and tangent components, where the radial is represented with spin $s = 0$ and the tangent has $s = 1$. Since the intersection distance is also a function with $s = 0$, our 3D shape representation has two spherical channels with $s = 0$ and one of $s = 1$. Following Cohen et al. [11], we also use the convex hull for extra channels.

When inputs have limited orientations, a globally equivariant model can be undesirable, even though equivariance in the local sense is still useful. We can keep the benefits while still having access to the global pose by breaking equivariance on the final layers, which we do by simply replacing them with regular 2D convolutions. We call this model "Ours + BE"; it results in better performance on "upright" but worse on "rotated", as expected.

Table 4 compares with previous spherical CNNs. The "upright" mode has only azimuthal rotations while "rotated" is unrestricted. EMVN [18] is state-of-the-art on this task with 94.4% accuracy on "upright" and 91.1% on "rotated", but it requires 60 images as input and much larger model.

## 5.5  Semantic segmentation of spherical panoramas

We evaluate our method on the Stanford 2D3DS dataset [2], following the usual protocol of reporting the average performance over the three official folds.

As in Section 5.4, our model is able uniquely represent surface normals. In this task, representing the normals with respect to local tangent frames is also more realistic, as they could be estimated from a depth sensor without knowledge of global orientation. Note that competing methods don't usually leverage the normals, so we also show results without them for comparison.

Table 5: Semantic segmentation on Stanford 2D3DS. Our model clearly outperforms previous equivariant models and matches the state-of-the-art non-equivariant model.

|  | acc [%] | mIoU |
|---|---|---|
| UGSCNN [24] | 54.7 | 38.3 |
| Gauge CNN [8] | 55.9 | 39.4 |
| HexRUNet [47] | 58.6 | 43.3 |
| SphCNN [16] | 52.8(6) | 40.2(3) |
| Ours | 55.6(5) | 41.9(5) |
| +normals | 57.5(6) | 43.4(4) |
| +normals+BE | 58.7(5) | 43.4(4) |

Table 5 shows the results. Inputs are upright so global **SO**(3) equivariance is not required; nevertheless, our method matches the state-of-the-art performance, which demonstrates the expressivity of the SWSCNNs.

## 6  Conclusion

In this paper, we introduced the spin-weighted spherical CNNs, which use sets of spin-weighted spherical functions as features and filters, and employ layers of a newly introduced spin-weighted spherical convolution to process spherical images or spherical vector fields. Our model achieves superior performance on the tasks attempted, at a reasonable computational cost. We foresee further applications of the SWSCNNs to 3D shape analysis, climate/atmospheric data analysis and other tasks where inputs or outputs can be represented as spherical images or vector fields.

## Broader Impact

This paper presents advances on learning representations from spherical data. It has potential beneficial applications to climate and atmospheric modeling, for example.

The method is in the broad category of equivariant CNNs, which have the goal to reduce model and sample complexity and improve generalization performance. This potentially translates to models that are more energy efficient, and are more accessible to individuals without access to large computational resources. On the flip side, most technology can also be applied for harmful purposes, and when making it more accessible we also risk enabling bad actors to make use of it.

## Acknowledgments and Disclosure of Funding

Research was sponsored by the Army Research Office and was accomplished under Grant Number W911NF-20-1-0080 as well as NSF TRIPODS 1934960 and the ONR N00014-17-1-2093 grants. The views and conclusions contained in this document are those of the authors and should not be interpreted as representing the official policies, either expressed or implied, of ARO, ONR, or the U.S. Government. The U.S. Government is authorized to reproduce and distribute reprints for Government purposes notwithstanding any copyright notation herein.

## Footnotes

[1]The subscripts $m$, $n$ refer to rows and columns of the matrix, respectively.

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
