[Supplementary Material]

# Supplementary Material
# Spin-Weighted Spherical CNNs

**Carlos Esteves**
GRASP Laboratory
University of Pennsylvania
machc@seas.upenn.edu

**Ameesh Makadia**
Google Research
makadia@google.com

**Kostas Daniilidis**
GRASP Laboratory
University of Pennsylvania
kostas@cis.upenn.edu

## 1 Introduction

In this supplementary material we give more details about the datasets in Section 2, about the experiments in Sections 3 to 5, and we describe the spin-weighted spherical harmonic (SWSH) transform implementation in Section 6.

## 2 Datasets

We show samples of the spherical vector field MNIST (SVFMNIST) dataset in Fig. 1. This is the dataset used in the vector field classification task.

Figure 1: Samples from SVFMNIST, classification task. We show one sample for each category in canonical orientation for easy visualization.

For the dense prediction tasks, we introduce modifications in the targets to make them more challenging. When predicting an image from a vector field, we introduce color in the output based on the target category. We determine the color in HSV space, where the value is the original grayscale value, the hue is $c/10$ for category $c$, and the saturation is set to one. The target is then converted back to RGB. Figure 2 shows a few input/target pairs.

When predicting a vector field from an image, we introduce an angular offset on all vectors that depends on the target category. The offset for category $c$ is given by $\exp(2\pi i c/10)$. Figure 3 shows a few input/target pairs.

Figure 2: Samples from SVFMNIST, image from vector field prediction task. Top shows input vector fields, bottom the target spherical images. Note that the targets have different colors based on the category, so the task cannot be solved via simple gradient integration. Samples are in canonical orientation for easy visualization.

Figure 3: Samples from SVFMNIST, vector field from image prediction task. Top shows input spherical images, bottom the target vector fields. The targets have different angular offsets based on the category so the task cannot be solved via simple image gradient estimation. Samples are in canonical orientation for easy visualization.

# 3 MNIST Experiments Details

In these experiments, we train for 12 epochs using the Adam optimizer [6]. We set the initial learning rate to $1 \times 10^{-3}$ and decay it to $2 \times 10^{-4}$ epoch 6 and $4 \times 10^{-5}$ at epoch 10. The mini-batch size is set to 32 and input resolution is $64 \times 64$.

The usual cross-entropy loss is optimized for the classification experiments, and the mean squared error is minimized for dense prediction.

## 3.1 Classification

The architectures for spherical image and vector field classification are the same.

The spherical baseline follows Esteves et al. [2], with spherical convolutions, six layers with $16, 16, 32, 32, 58, 58$ channels per layer, and 8 filter parameters per layer.

We follow the same general topology, switching from spherical to spin-weighted convolutions. Since our filters have richer spectra, they need more parameters. In order to keep similar number of parameters between competing models, we set the number parameters per spin-order pair $(s, m)$[1]

to $6, 6, 4, 4, 3, 3$ at each layer. We also cut the number of channels per layer, so while we have the same number of parameters, we have significantly fewer feature maps. The final architecture has $16, 16, 20, 24, 28, 32$ channels per layer, with pooling every two layers, and our custom batch normalization applied at every layer.

The planar baseline has the same number of layers and uses 2D convolutions with $3 \times 3$ kernels. We set the number of channels per layer to $16, 16, 32, 32, 54, 54$. to match the number of parameters of the other models.

## 3.2 Spherical vector field/image prediction

We design a different architecture for dense prediction, which is essentially a fully convolutional U-Net [9] with spin-weighted convolutions.

We use $16, 32, 32, 32, 32, 16$ channels per layer, with pooling in the first two layers and nearest neighbors upsampling in the last two. The number of filter parameters chosen per spin-order per layer is $6, 4, 3, 3, 4, 6$.

The spherical CNN baseline uses spherical convolutions and sets the numbers of filter parameters to 8 per layer and the number of channels to $20, 40, 78, 78, 40, 20$.

The planar baseline again uses 2D convolutions with $3 \times 3$ kernels and of channels to $18, 36, 72, 72, 36, 18$ channels.

## 3.3 Input-output samples

We show extra examples of inputs and outputs for the dense prediction tasks. Figure 4 shows the vector field to image task while Fig. 5 shows the image to vector field task. Models are trained on the R mode, so they have access to rotated samples at training time. Nevertheless, the standard CNN and spherical CNN models are not equivariant in the vector field sense and cannot achieve the same accuracy as the spin-weighted spherical CNNs (SWSCNNs).

| Input | CNN | SphCNN | Ours | GT |

Figure 4: Input/output samples for the spherical vector field to image task. We show two rotated instances of the same input to highlight that standard CNNs and spherical CNNs do not respect the spherical vector field equivariance, while the SWSCNNs do.

# 4 Classification of 3D shapes

ModelNet40 [11] training and test sets contain 9,843 and 2,468 CAD models, respectively. We evaluate following the protocol from Esteves et al. [2] that includes multiple rotated copies of each object in training and test sets. The "upright" mode has azimuthal rotations only, while the "rotated" mode has arbitrary 3D rotations.

Figure 5: Input/output samples for the spherical image to vector field task.

We train for 48 epochs using the Adam optimizer [6], with learning rate linearly increasing from 0 to $5 \times 10^{-3}$ during the first epoch then decayed by a factor of 5 at epochs 32 and 44. The mini-batch size is 32 and input resolution is $64 \times 64$. The cross-entropy loss is optimized and we found that label smoothing regularization [10] with $\epsilon = 0.2$ is beneficial.

The basic block is residual [3] with a bottleneck halving the number of channels when input and output have equal number of channels. Our custom batch normalization and nonlinearity is applied to the complex feature maps. We use $32, 32, 64, 64, 128, 128, 256, 256$ channels per layer where average pooling is applied before each increase in the number of channels, and $6, 6, 4, 4, 3, 3, 3, 3$ filter parameters are learned per spin-order per layer, with a total of $1.2\,\mathrm{M}$ parameters. When breaking equivariance in "Ours + BE", we replace the last two layers by three blocks of 2D convolution with $3 \times 3$ kernels.

The same training procedure and architecture are used for the SphCNN [2] baseline, which explains the superior numbers we report when comparing with the original paper.

We evaluate the baseline from Jiang et al. [5] following the recipe in the paper. The only difference is that we randomly rotate the training and test sets. Each training set object is rotated multiple times to serve as augmentation. The numbers we obtain differ from the $90.5\,\%$ accuracy reported in the original paper because our results are for azimuthally and arbitrarily rotated datasets while the original has all objects in a canonical pose.

## 5    Semantic segmentation of spherical panoramas

The Stanford 2D3DS dataset [1] contains 1,413 RGB-D panoramas with corresponding pixelwise semantic labels and normals. We follow the protocol from Jiang et al. [5] that reports pixelwise accuracy and mean intersection-over-union (mIoU) averaged over the three official folds. We also use the same weights per class as Jiang et al. [5] to mitigate the class imbalance.

We train for 48 epochs using the Adam optimizer [6], with the learning rate linearly increasing from 0 to $1 \times 10^{-2}$ during the first epoch then decayed by a factor of 10 at epoch 40. The mini-batch size is 8 and input resolution is $128 \times 128$. The pixelwise cross-entropy loss is optimized with label smoothing regularization [10] with $\epsilon = 0.2$.

A fully convolutional U-Net [9] architecture is used with same residual block described in Section 4. We use $16, 64, 128, 128, 256, 256, 128, 128, 64, 16$ channels per layer where average pooling/nearest neighbor upsampling is applied before each increase/decrease in the number of channels, and $8, 6, 6, 4, 4, 3, 3, 4, 4, 6, 6, 8$ filter parameters are learned per spin-order per layer, with a total of $2.5\,\mathrm{M}$ parameters. When breaking equivariance in "Ours + BE", we replace the last layer by six blocks of 2D convolutions with $3 \times 3$ kernels and 32 channels.

# 6   Spin-Weighted Spherical Harmonics Transforms

Our implementation of the SWSH decomposition and its inverse follows Huffenberger and Wandelt [4]. The basic idea is to leverage the relation between the SWSHs and the Wigner-D matrices. Recall that we can write the Wigner-D matrices as

$$D_{m,n}^{\ell}(\alpha, \beta, \gamma) = e^{-im\alpha} d_{m,n}^{\ell}(\beta) e^{-in\gamma}, \tag{1}$$

where $d^{\ell}$ is a Wigner-d matrix.

We define $\Delta_{m,n}^{\ell}$ as

$$\Delta_{m,n}^{\ell} = d_{m,n}^{\ell}(\pi/2), \tag{2}$$

then the following relation holds [8],

$$d_{m,n}^{\ell}(\theta) = i^{m-n} \sum_{k=-\ell}^{\ell} \Delta_{k,m}^{\ell} e^{-ik\theta} \Delta_{k,n}^{\ell}. \tag{3}$$

Now we rewrite the SWSH forward transform,

$$
\begin{aligned}
{}_s\hat{f}_m^{\ell} &= \int_{\theta,\phi} {}_sf(\theta,\phi)\overline{{}_sY_m^{\ell}(\theta,\phi)} \, \sin\theta \, d\theta \, d\phi \\
&= \int_{\theta,\phi} {}_sf(\theta,\phi)(-1)^s \sqrt{\frac{2\ell+1}{4\pi}} e^{is\psi} D_{m,-s}^{\ell}(\phi,\theta,\psi) \, \sin\theta \, d\theta \, d\phi \\
&= (-1)^s \sqrt{\frac{2\ell+1}{4\pi}} \int_{\theta,\phi} {}_sf(\theta,\phi) e^{-im\phi} d_{m,-s}^{\ell}(\theta) \, \sin\theta \, d\theta \, d\phi \\
&= (-1)^s \sqrt{\frac{2\ell+1}{4\pi}} \int_{\theta,\phi} {}_sf(\theta,\phi) e^{-im\phi} i^{m+s} \sum_{k=-\ell}^{\ell} \Delta_{k,m}^{\ell} e^{-ik\theta} \Delta_{k,-s}^{\ell} \, \sin\theta \, d\theta \, d\phi \\
&= (-1)^s i^{m+s} \sqrt{\frac{2\ell+1}{4\pi}} \sum_{k=-\ell}^{\ell} \Delta_{k,m}^{\ell} \Delta_{k,-s}^{\ell} \int_{\theta,\phi} {}_sf(\theta,\phi) e^{-im\phi} e^{-ik\theta} \, \sin\theta \, d\theta \, d\phi \\
&= (-1)^s i^{m+s} \sqrt{\frac{2\ell+1}{4\pi}} \sum_{k=-\ell}^{\ell} \Delta_{k,m}^{\ell} \Delta_{k,-s}^{\ell} I_{k,m}.
\end{aligned}
$$

Since the $\Delta_{m,n}^{\ell}$ are constants, they are pre-computed. We still need to compute

$$I_{k,m} = \int_{\theta,\phi} {}_sf(\theta,\phi) e^{-im\phi} e^{-ik\theta} \, \sin\theta \, d\theta \, d\phi, \tag{4}$$

which can be done efficiently with an FFT. There is a problem because ${}_sf$ is defined on the sphere so it is not periodic in both directions; we then define ${}_sf'$ as the periodic extension of ${}_sf$ which is a function on the torus. See McEwen [7] and Huffenberger and Wandelt [4] for more details about this extension. We can then express ${}_sf'$ by its Fourier coefficients,

$${}_sf'(\theta,\phi) = \sum_{p,q} {}_s\hat{f}_{p,q}' e^{ip\theta} e^{iq\phi}. \tag{5}$$

Substituting this in Eq. (4) yields,

$$I_{k,m} = \int\limits_{\theta=0}^{\pi} \int\limits_{\phi=0}^{2\pi} \sum_{p,q} {}_s\hat{f}'_{p,q} e^{ip\theta} e^{iq\phi} e^{-im\phi} e^{-ik\theta} \, \sin\theta \, d\theta \, d\phi$$

$$= \sum_{p,q} \int\limits_{\theta=0}^{\pi} \int\limits_{\phi=0}^{2\pi} {}_s\hat{f}'_{p,q} e^{i(p-k)\theta} e^{i(q-m)\phi} \, \sin\theta \, d\theta \, d\phi$$

$$= \sum_{p} 2\pi \int\limits_{0}^{\pi} {}_s\hat{f}'_{p,m} e^{i(p-k)\theta} \, \sin\theta \, d\theta$$

$$= 2\pi \sum_{p} {}_s\hat{f}'_{p,m} \hat{w}(p-k),$$

where $\hat{w}$ can be obtained analytically. Note that the last expression is a 1D discrete convolution; if we see $\hat{w}$ as the Fourier transform of some $w$, the convolution can be evaluated as the FFT of the multiplication in the spatial domain,

$$I_{k,m} = \frac{2\pi}{N^2} \sum_{\theta,\phi} {}_s f'(\theta,\phi) w(\theta) e^{-ik\theta} e^{-im\phi}, \tag{6}$$

for $N$ uniformly sampled $\theta$, $\phi$. Here, $w$ can be pre-computed, so the $I_{k,m}$ computation amounts to 1) extend the function to the torus, 2) apply the weights $w$, 3) compute a 2D FFT.

## Footnotes

[1]We use spins 0 and 1 throughout: $M_F = M_K = \{0, 1\}$. This amounts to four spin-order pairs per filter per degree: $_0k_0^\ell, \ _0k_1^\ell, \ _1k_0^\ell, \ _1k_1^\ell$.