[Reviews · NeurIPS 2020]

Review 1

Summary and Contributions: This paper presents a novel and efficient spherical CNN. The idea is to use spin-weight spherical functions that were introduced in physics for gravitational waves study. The new spherical CNNs are constructed with the novel convolution with spin-weighted functions, enabling both expressive and efficient CNNs on non-Euclidean domains. The spin-weighted spherical harmonics have applications in gravitational radiation and electromagnetic theory. They are generalizations of standard spherical harmonics (when setting $s$ to zero) and more expressive. To make it model rotation easily, the equivalent convolution definition from [5] is used. The proposed approach is novel but also a little overcomplicated. Updates: After reading the authors' feedback, I think my concerns are well-addressed. As such, I raised my score and vote for acceptance.

Strengths: (1) Leveraging spin-weighted spherical functions to build spherical CNNs is novel and interesting. The proposed approach can achieve the expressivity of SO(3) convolutions and keep the efficiency of CNNs between spherical functions. (2) Using spin-weighted spherical functions can achieve equivariance over vector fields. (3) Detailed and useful background introduction is given. (4) Experiments on spherical image classification, spherical vector field classification, and spherical vector field prediction show promising performances and efficiency. (5) Examples are provided for intuitive explanation.

Weaknesses: The proposed approach is slower than [15] because of the forward and backward Fourier transforms. This is acceptable though. The running time of [7] is not shown in Table 1. Source code is not provided. I am wondering if you could show us some results on the same tasks shown in [15], such as results on 3d object classification (ModelNet40) and 3D object retrieval ( ShapeNet Core55), making them comparable.

Correctness: I did not make a full check on the maths.

Clarity: The paper is well written and organized. It is easy to follow overall but I think it would be better to give more intuitive explanations.

Relation to Prior Work: Relations to previous contributions such as equivariant CNNs, spherical CNNs, and equivariant vector fields are discussed. Here are some missing related works: SPHERICAL CNNS ON UNSTRUCTURED GRIDS (ICLR 2019)

Reproducibility: Yes

Additional Feedback: 1. Are there possible solutions to further improve the efficiency to match that of [15]? The authors claim the efficiency of the proposed approach several times, but from the complexity analysis, the approach is slower than [15], right? 2. The running time of some baselines is missing. 3. In equation (2), the number of band is usually limited, right? So it is not infinite sum. 4. The authors use the equivariant convolutions definition (12) from Boyle, but this definition is not true when transformation is more complicated as stated in [5]. Will it bring any issues to the model? 5. I also expect some results on the same tasks as shown in [15]. 6. What are the benefits of spin basis compared to Wigner D-matrices?


Review 2

Summary and Contributions: This work designed a spin-weighted Spherical CNN kernel with several properties: has more expressive representations than scalar CNN. enable anisotropic filters. friendly for vector field-related tasks.

Strengths: Define a novel way for convolution that handles vector fields, and outperform existing methods on traditional tasks.

Weaknesses: It is hard for me to fully understand the mathematics. I feel that authors should expose more details on the part of SWSH. If space is not enough, consider shorten the spherical harmonics since that's more popular to readers. Many variable/notations are just written without clear definition, which makes readers hard to get access to. From my understanding, the authors define a new and powerful kernel for rotationally equivariant CNNs. However, rotation equivariance is not a good property for many data, e.g., spherical image is usually with a fixed up/gravity vector. Vector fields are commonly studied on sphere like the earth where poles are clearly defined. These issues are properly addressed by "Jiang, Chiyu, et al. "Spherical CNNs on unstructured grids." arXiv preprint arXiv:1901.02039 (2019)." However, the important related work is missing without comparison. Another weakness is again about the experiment. It only plays with MNIST, which is insufficient demonstrate the usefulness of the methods. Therefore, I highly recommend the authors to perform experiments on real data (like what "Spherical CNNs on unstructured grids.", and compare with them).

Correctness: Seemingly correct after I spend effort checking mathematics background.

Clarity: The introduction is fine. The fundamental mathematics are too brief for readers to understand.

Relation to Prior Work: Missing related works: 1. Jiang, Chiyu, et al. "Spherical CNNs on unstructured grids." arXiv preprint arXiv:1901.02039 (2019). 2. Huang, Jingwei, et al. "Texturenet: Consistent local parametrizations for learning from high-resolution signals on meshes." Proceedings of the IEEE Conference on Computer Vision and Pattern Recognition. 2019.

Reproducibility: Yes

Additional Feedback: The authors mentioned climate data. Please provide experiments (check Jiang et al.) to make it more convincing.


Review 3

Summary and Contributions: Recent work on spherical CNNs can be broadly classified into two categories: One that lifts the functions on the sphere (both the input image and the filter) onto the group SO(3), where the convolution is performed. Another that operates fully on the sphere, however such networks don't allow for anisotropic filters and are thus of limited expressivity. The main contribution of this paper is to propose spherical CNNs that allow for anistropic filters, never leave the spherical domain and thus are much more efficient to implement while giving good experimental performance. Towards this end, the authors employ spin-weighted spherical harmonics (a generalization of spherical harmonics, that suffer a phase change when the sphere is subject to a rotation) for performing the convolution in the spectral domain. The SWSF's allow for more expressive representations than the usual scalar spherical functions. In fact SWSFs are gauge fields which take values in a line bundle over the sphere. To give a complete specification for the CNN, analogs for non-linearities, batch-normalization and pooling layers are defined. Finally, as mentioned above, since the SWSFs can be interpreted as equivariant vector fields, the method proposed is useful when the input/outputs are vector fields. One could think of the proposed method as the analog of "Harmonic Networks" of Worrall et al, but where the functions are now defined on a sphere. Section 3 covers the background. First covering the usual spherical harmonics formulation, followed by Spin-Weighted spherical harmonics. From the formulation it can also be seen that that SWSHs can be seen as functions on the rotation group with a sparse spectrum. This also gives a good intuition on how the proposed method lies somewhere in between the two approaches described above. Section 4 defines the usual notion of cross-correlation but now using the SWSHs and shows clearly its equivariance properties. The following section provides an analog of batch norm, gives the non-linearities that are equivariant. The experiments results compare to the existing Spherical CNNs and exhibits superior performance. In summary the main contribution of the paper is to write the spherical convolution using the SWSHs, allowing for working with inputs and outputs that are vector fields, anisotropic filters, and always operating on the sphere. The results are improved accuracy and reduced computational costs.

Strengths: The main claim of the paper is that using the spin-weighted spherical harmonics in place of the spherical harmonics allows for modeling more powerful spherical CNNs. In particular as they allow for using anisotropic filters and always operating on the sphere, thus also providing a more efficient spherical CNN. In the above sense the contribution of the paper is only a change in how the convolution in spherical CNNs is performed. However the formulation is elegant and gives an immediate boost in performance.

Weaknesses: There are three sets of experiments in the paper each of which operates on variations on a spherical MNIST. This can be seen as a negative. As in this case it is not possible to fully infer the computational benefits that are used as a motivation in the beginning. Some examples of non-trivial data would include: CMB (Cosmic Microwave Background) data, predicting polarization (since that is a vector field) or publicly available geophysical data that involve input data that are magnetic fields over the earth.

Correctness: Yes.

Clarity: Yes overall it is well written. Some minor comments: Section 2: Line 75: "Cohen and Welling [11] formalize these models and name them group convolutional neural networks (G-CNNs)". It would be preferable to write "formalized these models and named them..". Same on page 96: "Cohen et al. introduce.."

Relation to Prior Work: Relation to prior work is well documented. One suggestion: The author might want to cite https://openreview.net/forum?id=HJeYSxHFDS this also allows for anisotropic filters on the spheres, and is gauge equivariant. However this work is not online on arXiv and does not have a doi. Moreover, openreview is not indexed. Therefore, it is entirely on the author's discretion if they would like to cite it. I wanted to bring this to their notice.

Reproducibility: Yes

Additional Feedback: Update after feedback: Thanks to the authors for addressing some of the concerns raised. The additional experiments are welcome and will strengthen the paper.


Review 4

Summary and Contributions: This paper introduced the spin-weighted spherical CNNs, a new type of spherical CNN that allows anisotropic filters in an efficient way, without ever leaving the spherical domain. The key idea is to consider spin-weighted spherical functions, which were introduced in physics in the study of gravitational waves. These are complex-valued functions on the sphere whose phases change upon rotation. It uses sets of spin-weighted spherical functions as features and filters, and employ layers of a newly introduced spin-weighted spherical convolution to process spherical images or spherical vector fields. Experiments show that the proposed method outperforms the isotropic spherical CNNs while still being much more efficient than using SO(3) convolutions. Our model achieves superior performance on the tasks attempted, at a reasonable computational cost.

Strengths: Overall this is a strong paper with very solid technical foundation. Although the idea is originally from physics but its application in computer vision is novel. The performance seems outperform existing state of arts

Weaknesses: As a reviewer not very familiar with the technical part, it seems hard to understand all the mathematical formulas in the paper. It would be better if the paper can provide more self contained description so that ordinary user can follow better. For example in Table 3, NR, R is not explained. Also the three examples given in the paper seems relatively idealized. It would be better if it can show some more real applications. I do understand this paper is more focused on the theory foundations.

Correctness: Yes

Clarity: Yes

Relation to Prior Work: Yes

Reproducibility: Yes

Additional Feedback: I will keep my ratings after the rebuttal.

[Author Response · NeurIPS 2020]

We thank the reviewers for constructive comments and suggestions. Reiterating our contributions: we introduce the
spin-weighted spherical CNNs to strike a balance between expressivity and efficiency in the context of equivariant
spherical CNNs. Our goal is to be more efficient than the SO(3)-based models introduced by Cohen et al. [7] and more
expressive than the purely spherical introduced by Esteves et al. [15]. We achieve it with a computation complexity
closer to the more efficient model, while outperforming both models. Moreover, we make a fundamental contribution
on equivariant processing of spherical vector fields that enables future scientific applications as suggested by **R3**.

Although reviewers recognized our approach as novel,**R1, R2, R4** interesting,**R1** elegant,**R3** and its promising
performance,**R1, R2, R3, R4** they all suggested extra experiments. While we believe our claims are adequately sup-
ported, we agree that new experiments will strengthen the paper. We evaluate our model for 3D shape classification**R1**
and panoramic image segmentation**R2** and show results in Tables 1 and 2. The classification model takes simpler
inputs (4k pixels, two channels instead of the 15k+ pixels, six channels of Jiang et al.[1] and Cohen et al [7]), and a
simpler architecture (single branch instead of the two branches of Esteves et al [15]). The performance is competitive
nevertheless. One potential improvement that is unique to our model is that we could represent surface normals with
spin $s = 0$ and $s = 1$ components, achieving more faithful input representation without sacrificing the equivariance.

|  | Esteves [15] | Jiang[1] | Ours | Ours + BE |
|---|---|---|---|---|
| mIoU | 0.363 | 0.383 | 0.398 | 0.421 |

Table 1: Stanford 2D3DS spherical panorama segmentation.

|  | Esteves [15] | Jiang[1] | Ours | Ours + BE |
|---|---|---|---|---|
| aligned | 88.1 | 90.5 | 88.5 | 89.4 |
| rotated | 86.9 | - | 87.2 | 86.8 |

Table 2: ModelNet40 shape classification accuracy [%].

**R2** - *"rotation equivariance is not a good property for many data"*. While some datasets are guaranteed to be upright,
obviating the need for a globally equivariant model, the benefits of (local) equivariance in terms of filter sharing
and efficient use of network capacity still apply. This effect has been demonstrated on CIFAR10/100 by Cohen and
Welling [8] and more recently by Weiler and Cesa [36]. Moreover, it is trivial to break equivariance at some layer of
an equivariant model, while it is not possible to transform a non-equivariant feature map into an equivariant one. We
demonstrate this with the "Breaking-Equivariance" (BE) variants of our model in Tables 1 and 2. We simply add a
few standard 2D convolutional layers after the last equivariant feature map. This effectively breaks equivariance so the
model can leverage global orientation. As expected, it helps on aligned datasets but not on rotated.

**R1, R2** - *"citation and comparison with Jiang et al.[1]"*. This was an oversight of our part; we will include the citations
and comparisons elaborating on Tables 1 and 2. Note that the model in Jiang et al[1] is not equivariant to rotations and
hence is expected to underperform on rotated datasets, in contrast with ours.

**R1** - *"Source code is not provided"*. As mentioned in the introduction, the code and datasets will be released.

**R1** - *"benefits of spin basis compared to Wigner D-matrices?"*. The benefits are efficiency and flexibility. The spin
basis is related to sets of columns of the Wigner D-matrices. We can choose the number of spins per layer which will
define the size of the basis, enabling selection of the desired trade-off between expressivity and efficiency.

**R1** - *"further improve the efficiency to match that of [15]"*. Asymptotically, the complexity of Esteves et al [15] is
$\mathcal{O}(B^2 \log^2 B)$ while ours is $\mathcal{O}(B^3)$. In practice, this is acceptable in exchange for the expressivity improvements and
applicability to a wider range of tasks, while still comparing favorably against the $\mathcal{O}(B^4)$ of Cohen et al [7].

**R1** - *"it is not infinite sum"*. In practice, for computation, we need to limit the bandwidth. What equation (2) shows is
that, in theory, any square-integrable function on the sphere can be expanded as the infinite sum. This is analogous to
the Fourier series of functions on a real interval – it may need infinite terms but in practice we compute a finite number.

**R1** - *"when transformation is more complicated ... issues to the model?"*. If inputs are perturbed by transformations
that are not rotations, our model needs to learn them from data. This is no different than how standard neural networks
operate. If the set of perturbations also include rotations, our model still has the equivariance advantage.

**R2, R3** - *"applications to climate, cosmological, geophysical data"*. We truly appreciate the suggestions for possible
applications and plan to address them in future work, especially regarding processing of vector fields on the sphere.

**R1, R2, R4** - *"overcomplicated, hard to understand the math"*. We believe the prerequisites are at same level of Cohen
et al [7] and Esteves et al [15], and we list references to books and papers with the necessary harmonic analysis and
spin-weighted functions background. Nevertheless, we will revise the text for clarity and from a didactic standpoint.

**All** - We thank the reviewers for catching typos, missing data and missing citations. We will update accordingly.

## Footnotes

[1] Jiang et al, Spherical CNNs on Unstructured Grids. ICLR'19.


[Meta-Review · NeurIPS 2020]

This paper introduces a new type of spherical CNN based on spin-weight spherical functions that were introduced in physics for gravitational waves study. The exploited spin-weighted spherical harmonics, as generalizations of standard spherical harmonics, are more expressive and allow for working with inputs and outputs that are vector fields. Good experimental results illustrate the advantage of the proposed spherical CNN over traditional ones. The new experimental results added in the authors' response nicely addressed reviewers' concerns and should be included in the paper.